# Quo Vadis? Immunodynamics of Myeloid Cells after Myocardial Infarction

**DOI:** 10.3390/ijms232415814

**Published:** 2022-12-13

**Authors:** Aldo Moggio, Heribert Schunkert, Thorsten Kessler, Hendrik B. Sager

**Affiliations:** 1Department of Cardiology, German Heart Center Munich, Technical University Munich, 80636 Munich, Germany; 2DZHK (German Centre for Cardiovascular Research), Partner Site Munich Heart Alliance, 80336 Munich, Germany

**Keywords:** myocardial infarction, hematopoiesis, myeloid cells, extracellular signaling, inflammation, cellular heterogeneity

## Abstract

Myocardial infarction (MI), a major contributor to worldwide morbidity and mortality, is caused by a lack of blood flow to the heart. Affected heart tissue becomes ischemic due to deficiency of blood perfusion and oxygen delivery. In case sufficient blood flow cannot be timely restored, cardiac injury with necrosis occurs. The ischemic/necrotic area induces a systemic inflammatory response and hundreds of thousands of leukocytes are recruited from the blood to the injured heart. The blood pool of leukocytes is rapidly depleted and urgent re-supply of these cells is needed. Myeloid cells are generated in the bone marrow (BM) and spleen, released into the blood, travel to sites of need, extravasate and accumulate inside tissues to accomplish various functions. In this review we focus on the “leukocyte supply chain” and will separately evaluate different myeloid cell compartments (BM, spleen, blood, heart) in steady state and after MI. Moreover, we highlight the local and systemic kinetics of extracellular factors, chemokines and danger signals involved in the regulation of production/generation, release, transportation, uptake, and activation of myeloid cells during the inflammatory phase of MI.

## 1. Introduction

Cardiovascular diseases (CVDs) are the leading cause of global death. WHO estimations show that 17.9 million people died from CVDs in 2019, representing 32% of worldwide deaths. Of these, 85% were caused by myocardial infarction (MI) and stroke, both major complications of atherosclerosis [1,2]. Indeed, the most frequent cause of myocardial infarction is the progression and destabilization of atherosclerotic plaques and the formation of occlusive thrombi [3]. There are several classical risk factors for atherosclerosis initiation and progression such as smoking, dyslipidemia, diabetes mellitus, hypertension, obesity, but also non-classical risk factors including mental stress, environmental factors (air and light pollution, noise), clonal hematopoiesis of indeterminate potential (CHIP), sedentary lifestyle and prior or concomitant inflammatory diseases [4,5,6,7,8] (Table 1 and Figure 1).

Nowadays, preventive and interventional measures have made tremendous steps to improve the prognosis after MI, but nonetheless MI-related mortality and morbidity remains high. MI is characterized by cardiac ischemia which may lead to cardiac injury with tissue death. Indeed, obstruction of coronary arteries causes deficient blood supply which triggers a decrease in nutrients and oxygen. As a consequence, cardiomyocytes (CMs) and surrounding tissue become hypoxic and necrotic. MI is characterized by the release of factors into the extracellular milieu by dying cells and the accumulation of cellular debris, which both trigger an intense immunological response. High numbers of leukocytes are recalled from the blood and reach the ischemic area, where they start to remove cellular debris and degrade extracellular matrix. This process is known as the inflammatory phase of MI [22,23]. The main protagonists of this phase are neutrophils and monocytes/macrophages which act as the “emergency crew”. Neutrophils, the most abundant granulocytes in mouse blood, are characterized by a specific nuclear morphology, defined granule content and expression of the membrane markers CD11b and Ly6G, while CD11b and CD66b characterize human neutrophils [24]. Due to their limited life span (<24 h), the bone marrow (BM) provides a continuous supply of newly-made neutrophils, but MI or other types of stressors can cause accelerated “emergency” granulopoiesis [25]. Monocytes are circulating leukocytes known to give rise to macrophages after tissue invasion. BM and circulating monocytes can be identified based on the expression of CD11b and CD115 (macrophage colony-stimulating factor 1 receptor, MCSF1R). The expression level of Ly6C classifies monocytes into distinct subclasses: classical or pro-inflammatory (Ly6c^HI^), intermediate (Ly6c^INT^) and non-classical (Ly6c^LOW^) monocytes. Ly6c^HI^ monocytes (described as CD14^+^CD16^−^ in human blood monocytes) are the most abundant among circulating monocytes. In the steady state, the BM and spleen represent the main reservoirs of classical monocytes. In case of tissue damage or infection, they are released in the blood, reach the damaged tissue and give rise to pro-inflammatory macrophages upon tissue invasion [26]. Non-classical Ly6c^LOW^ monocytes (CD14^LOW^CD16^+^ in human blood monocytes) are characterized by the expression of C-X3-C motif chemokine receptor 1 (CX3CR1) and patrol the vasculature to check the integrity of the endothelial walls. Ly6c^INT^ monocytes (CD14^+^CD16^+^ in humans) represent a phenotypical bridge between classical and non-classical monocytes [27,28].

Early after MI (already 12 h post-MI), a massive quantity of neutrophils is recruited to the site of injury. Neutrophil counts reach a peak within the myocardium at 24–48 h after infarction. Within the cardiac tissue, neutrophils act as phagocytes removing cell debris, extracellular matrix and dying cells. Furthermore, they stimulate the recruitment of blood monocytes by releasing extracellular factors [29]. After the first day, the number of pro-inflammatory monocytes start to increase exponentially until 3–4 days post-MI. Once inside the ischemic area, monocytes give rise to macrophages, which remove the cellular debris and extracellular matrix. The number of pro-inflammatory macrophages inside the infarcted heart peaks between day 3 and 4 [30,31,32]. The inflammatory phase, which lasts around 4 days, is the first of three stages that characterize cardiac injury and repair secondary to MI, and it is followed by the regenerative and the maturation phases [33,34] (Figure 2).

The regenerative phase occurs between 4- and 10-days post-MI. In the ischemic area, recruited macrophages change polarization from pro-inflammatory to anti-inflammatory/reparative (formerly defined as M1 and M2, respectively). Reparative macrophages release pro-fibrotic and anti-inflammatory factors such as vascular endothelial growth factor (VEGF), transforming growth factor β (TGFβ) and interleukin- (IL-) 4 and IL-10 which start the revascularization and the deposition of extracellular matrix allowing the repopulation of the ischemic area by endothelial cells and myofibroblast. Finally, during the maturation phase (from day 10 on) scar formation takes place. Myofibroblasts become quiescent, collagen filaments cross-link and the re-vascularization process ends [35,36] (Figure 2).

The development of murine models of MI helped to appreciate several pathological and immunological aspects of cardiac injury that were not known before. Permanent ligation of the left anterior descending (LAD) coronary artery or its temporary occlusion causing ischemia and reperfusion injury (IRI) represent the two main animal models to study MI. From a cellular point of view, these two methods differ in the type of injury. In case of permanent ligation, extreme and prolonged hypoxia induces blockage of aerobic metabolism leading to activation of apoptotic pathways in cardiomyocytes (CM). In case of IRI, the temporary hypoxia—followed by reperfusion—induces production of free radicals by CM and variation of their cellular pH which causes mitochondria swelling and necrotic cell death [37,38]. Comparing these two models, permanent ligation causes higher mortality, bigger infarct sizes and a more prominent drop of left ventricular metabolic volume, as revealed by PET imaging. Moreover, different studies showed that the time of day (circadian control) of the ligation affects the progression of the injury in case of permanent ligation, but not in IRI [39,40,41]. These distinctive aspects of these two surgical techniques also impact the immune response that is triggered by the injury. Indeed, the comparison of these two LAD ligation models showed that the quantity of accumulated leukocytes in the infarct area is increased in case of permanent occlusion, while the number of circulating monocytes did not differ between the two surgical strategies [42]. Similarly, another study showed that reperfusion reduced the total number of leukocytes recruited to the infarct area and impacted the temporal course of innate and adaptive immune responses. In particular, reperfusion shifts the peak of the innate immune response towards earlier time points and also dampens the adaptive immune response [43].

Research of the past decades established a dominant contribution of inflammation to all stages of MI [44]. However, due to the complexity of the processes, many features of the immunopathology of MI are still unclear. Indeed, the inflammatory phase is a multifaceted process involving different tissues, extracellular signaling and modifications of cellular phenotypes [45]. Thus, the comprehension of all different interactions and biological processes is fundamental in the search of specific therapeutical targets to improve MI-related morbidity and mortality.

The following review will cover different aspects of the inflammatory phase following MI. Specifically, we will focus on the immunodynamics of neutrophils, monocytes, and macrophages by reviewing different compartments separately. We will summarize:(1)Production of leukocytes in the **bone marrow** and **spleen**;(2)Release of leukocytes from the bone marrow and spleen into the **blood**;(3)Recruitment of leukocytes from the blood to the **heart**;(4)Actions of leukocytes within the **heart**;in the setting of acute myocardial infarction.

## 2. Bone Marrow Niches and Hematopoiesis

### 2.1. Hematopoiesis: The Generation of Blood Cells

Recent technological developments, including single cell-RNA sequencing (sc-RNAseq) and cell tracing technologies, allow a better understanding of the generation of mature blood cells and redefined the hematopoietic model. These technical advances allow to follow stem cells and their progeny at single cell level revealing novel gene dynamics during differentiation and highlighting different maturation stages [46,47]. Moreover, the combination of these technologies allowed a new interpretation of the hematopoietic system beyond the classical hierarchic tree, in which the cells advance from one stage to the next making binary decisions. Indeed, the actual model takes into account the heterogeneity within different stages of maturation, which is considered a continuum from one stage to another with the possibility of multiple trajectories. Accordingly, different studies support the idea of a diffusion map with intermixing between different transition steps [27,48,49,50,51].

Hematopoietic stem cells (HSCs) of the BM have been identified by the expression of the signaling lymphocytic activation molecule (SLAM) receptors (CD150^+^ CD48^−^), which distinguish them from other progenitors within the BM [52]. Lately, the analysis of the transcriptome demonstrated that hepatic leukemia factor (HLF) transcription factor (TF), or PAR bZIP family member, was the only TF highly enriched in HSCs in comparison with the other hematopoietic cells [50]. HSCs are the starting point of hematopoietic maturation and can, due to external or internal signals, give rise to multipotent precursors (MPPs). MPPs (CD150^−^ CD48^−^) have higher proliferative potential compared to HSCs, and are classified in various subtypes based on their biased capability to generate myeloid or lymphoid cells. Indeed, MPP2s and MPP3s are primed towards myeloid lineages and MPP4s are lymphoid-biased [52,53]. Recently, sc-RNAseq data showed that MMPs have mainly two programs: an erythroid (expressing Car1, Apoe, Gata2) or a joint lymphoid–myeloid (expressing, Flt3, Cd34, Igh, Dntt, Cd52, Mpo) program [50]. The transcriptome data of lymphoid–myeloid progenitors showed that the Ly6D^+^SiglecH^−^CD11c^−^ fraction is lymphoid-restricted and exhibits a strong B-cell potential (expressing Cd79a, Cd79b, VpreB1/2, Igll1, Ebf1, Pax5, or Blnk), while the Ly6D^−^SiglecH^−^CD11c^−^ subpopulation showed a mixed lymphoid-myeloid potential [54]. Following the maturation path, MPPs give rise to intermediate precursors like common myeloid progenitors (CMPs) and common lymphocyte precursors (CLPs). Regarding the lymphoid compartment, CLPs mature into Nk cells, T cells, B cells and plasmacytoid dendritic cells. The granulocyte macrophage precursors (GMPs), which derive from CMPs, produce both the granulocyte lineage and monocyte lineage. The maturation of granulocytes involves granulocyte progenitors (GPs), which then produce neutrophils, eosinophils and basophils. Similar to GMPs, monocyte-macrophage/dendritic cell precursors (MDPs) can produce monocytes and also dendritic cells (DCs), but not granulocytes. In particular, MDPs can give rise to common dendritic cell precursors (CDPs), which maturate into conventional DCs (cDCs). Ly6C^HI^ monocytes originate from a precursor with a strict monocyte differentiation potential named common monocyte precursors (cMoPs). Previous reports have shown that cMoPs derived from MDPs, however, a recent study challenged this view. Using a fate-mapping model (Ms4a3^Cre^-Rosa^TdT^), the authors showed that cMoPs derived from GMPs. Further studies will clarify this controversy and will uncover the development of cMoPs [27,55,56,57] (Figure 3).

### 2.2. Bone Marrow Niche: Where Everything Starts

The BM is the tissue responsible for the production of blood cells, a process called hematopoiesis. Through asymmetric divisions, HSCs maintain the stem cell pool and give rise to lineage-restricted progenitors, which in turn differentiate into more mature cells [58]. The coordination of stem cell maintenance and their maturation into the blood cells requires a complex system of communication between different cell types, enclosed in a microenvironment called niche [59]. Hematopoietic niches are located in the perivascular space of the BM and are associated with arterioles or sinusoidal blood vessels (the capillaries of the BM). Together with HSCs and endothelial cells (ECs), the niches harbor different other cell types: mesenchymal stem and stromal cells (MSC), adipocytes, osteoblasts and leukocytes. Additionally, it has been shown that sympathetic nerves localize close to HSCs [60,61]. The role of hematopoietic and non-hematopoietic niche components is to release extracellular factors or express membrane associated proteins that regulate maintenance (quiescence state), proliferation, retention or differentiation of HSCs [62] (Figure 4).

The key signals for the maintenance of HSCs are the stem cell factor (SCF or Kitl) and the CXC-chemokine ligand 12 (CXCL12 or stromal cell–derived factor, SDF). SCF binds the receptor KIT, while CXCL12 promotes HSC maintenance and retention by binding CXC-chemokine receptor 4 (CXCR4) [62]. Experiments with Cxcl12 conditional knockout mice (CAGG^CreER^;Cxcl12^F/−^) showed that, in the physiological state, loss of Cxcl12 led to an increased progenitor cell pool, together with a reduction of long-term HSCs [63]. On the other hand, conditional deletion of SCF, from both ECs and stromal cells, in Tie2^cre^;Lepr^cre^;Scf^Δ/FL^ mice, reduced its gene expression in the BM substantially compared to wild type (WT) mice, indicating that these two cell types are the major sources of SCF in the BM. Moreover, Tie2^cre^;Lepr^cre^;Scf^Δ/FL^ mice showed a significant reduction of the absolute numbers of HSCs compared to the control mice [64].

Based on their anatomical location, bone marrow ECs can be divided in sinusoidal and arteriolar ECs. Sinusoidal EC are characterized by the high expression of Flt4 (Vegfr-3) and low expression of Ly6a (Sca-1), while the arteriolar EC show the opposite pattern (Vegfr-3^low^/Sca-1^high^) [65]. Moreover, sinusoidal and arteriolar ECs can be distinguished using a combination of podoplanin (PDPN) and Sca-1 expression: arteriolar EC are Sca-1^bright^/PDPN^−^, while sinusoidal EC are Sca-1^dim^/PDPN^+^. Cxcl12 and Scf are mostly expressed by arteriolar ECs [66]. Further, it has been shown that ECs are the main source of additional factors that affect HSCs: angiogenin (Ang), selectin E (Sele) and delta-like NOTCH ligands (Dll4 and Dll1). NOTCH signaling, in particular, is fundamental for hematopoiesis and HSC self-renewal [67,68].

Perivascular MSCs are stromal cells with the ability to self-regenerate or to differentiate into osteocytes, chondrocytes or adipocytes. Furthermore, they have the fundamental role within the niche of modulating the maintenance of HSCs. MSCs have been identified and extensively studied by the expression of Nestin (Nes). Nes^+^ MSCs are associated with sympathetic nerves and are in contact with HSCs. They express genes that induce HSCs maintenance like Cxcl12, Angpt1, Il7, Vcam1 and Spp1 [69].

Analyzing CXCL12-GFP mice, it was shown that perivascular Cxcl12- abundant reticular (CAR) cells represent the major producers of CXCL12 within the BM. CAR cells show mesenchymal properties and are in direct contact to HSCs [70]. Using different mouse models of conditional ablation of Cxcl12 and Scf, other markers for identifying perivascular cells were discovered: leptin receptor (Lepr) and Nerve/glial antigen 2 (NG2). However, around 90% of Lepr+ cells overlap with CAR cells and represent a large subset (approximately 80%) of Nes^+^ cells [71]. A recent report on BM single cell-RNA sequencing (sc-RNAseq) analysis showed that Cxcl12 and Scf are mainly produced by two distinct Lepr^+^ subpopulations not characterized before. They have been named Adipo-CAR and Osteo-CAR since they exhibit an adipocyte (Adipoq^+^, Ldl^+^) and osteo-lineage (Sp7^+^) expression profile, respectively. The spatial analysis of the BM showed that Adipo-CAR, which expressed higher level of Lepr compared to osteo-CAR, are mostly peri-sinusoidal, while Osteo-CAR preferentially localize to arteriolar or non-vascular niches [72]. The spatial analysis confirmed a study from Kunisaki and colleagues, which showed that arterioles are interacting with NG2 pericytes differently from sinusoid-associated Lepr^+^ cells [73].

BM adipocytes affect HSCs proliferation and differentiation by secreting adiponectin, leptin, prostaglandins, IL-6, and other factors. Adiponectin induces proliferation of HSCs and maintains their undifferentiated state. On the other hand, IL-6 and leptins promote differentiation of HSCs, while prostaglandins inhibit HSCs through modulation of survival/apoptosis signaling [74]. Additionally, it has been shown that BM adipocytes express a high level of Scf which promotes regeneration of HSCs and hematopoiesis after irradiation or 5-fluorouracil (5-FU) treatment [75].

Osteoblasts represent another non-hematopoietic component of the niche. Contrary to other cell types, osteoblast do not modulate the quiescence of the HSC by Cxcl12 and Scf expression. Osteoblasts are the main source of Osteopontin (Opn), which showed an expression highly restricted to the endosteum of the bone marrow cavity. From a functional point of view, Opn is a negative regulator of HSC proliferation. Indeed, the deletion of Opn induces an increase of cycling HSCs [76].

Autonomic nerves associated to bones, spread into the BM. Out of all autonomic nerves reaching bones, around 5% of sympathetic nervous system (SNS) fibers actually penetrate the BM and can regulate hematopoiesis [77]. Indeed, HSCs are released into the blood in a circadian manner in response to adrenergic signals from the SNS. Specifically, noradrenaline, released by SNS, down-regulates the expression of Cxcl12 in perivascular cells (expressing β3-adrenergic receptor, β3AR) inducing egress of HSCs [78]. Among different cells within the hematopoietic niche, glial cells surround blood vessel walls. Non-myelinating Schwann cells may release active TGFβ, which triggers Smad signaling causing quiescence in HSCs [79].

Among cells that influence the dynamics of HSCs, other hematopoietic cells, such as megakaryocytes (MK), macrophages and T cells, need to be considered. Through the use of two different inducible diphtheria toxin receptor (iDTR) mouse models, two independent studies showed the fundamental role of MK in regulating the quiescence of HSCs. MKs are in direct contact with HSCs and through expression of CXCL4 and TGFβ negatively regulate proliferation of HSCs [80,81]. With a similar approach using CD169^DTR^ mice, Chow and colleagues showed that the depletion of macrophages induced HSC mobilization. The retention of HSCs in the BM relies on an interaction between macrophages and Nes^+^ cells, which antagonizes the mobilization signaling of the SNS [82]. Further analyses are needed to understand how macrophages and perivascular cells communicate. A population of FoxP3^+^ regulatory T cells (Treg) has been identified in the hematopoietic niche, but not in direct contact with HSCs. The Treg population, which is characterized by the expression of CD150, showed to regulate the maintenance of HSC quiescence and engraftment by the release of adenosine [83].

In summary, the hematopoietic niche is a complex system, tightly regulated by different factors allowing communications between the different cellular components (Figure 4). As we will illustrate, stressors, such as MI, can modify hematopoiesis through direct and indirect actions [84,85,86].

## 3. Myocardial Infarction Guided Emergency Hematopoiesis

While hematopoiesis continuously supplies new blood cells in homeostatic conditions, hematopoiesis may react to external stressors/triggers. The response of the hematopoietic system is driven by several signals, either local or external. In case of MI, cardiac cells (such as mast cells, macrophages, and CM) trigger a massive recruitment of circulating leukocytes through the release of soluble factors. In parallel, the BM responds to the remote injury by releasing hematopoietic progenitors, expanding the immune cell pool and promoting the maturation of distinct leukocyte subsets [44]. To meet the high demand of leukocytes, hematopoiesis may also take place outside the BM, in the spleen (extramedullary hematopoiesis) [27,87,88,89,90]. “Emergency hematopoiesis” occurs after MI and includes different processes: (1) the amplification of leukocyte production, (2) the maturation of alternative leukocyte subsets and (3) the release of myeloid cells from primary and secondary hematopoietic organs.

### 3.1. Progenitor Amplification

The BM responds to MI and transiently increases production of leukocytes. This process passes through the amplification of hematopoietic progenitor cells in the BM. Indeed, positron emission tomography (PET) analysis of the pelvic bone in patients with MI showed increased metabolic activities in the BM, probably reflecting increased proliferation of HSCs. In line, the analysis of murine models showed that MI drives the activation of Wnt pathway, which regulates the proliferation of hematopoietic stem and progenitor cells (HSPCs) [91]. In a study by Dutta and colleagues, it was shown that MI supports the proliferation of CCR2^+^ HSCs. Through the expression of Mtg16, CCR2^+^ progenitors are biased toward a myeloid phenotype, thus present a source of neutrophils, monocytes and macrophages, the cellular protagonists of the inflammatory phase following MI [84]. Using parabiosis, mice with a joined circulation, and LAD coronary artery ligation, the same research group demonstrated that MI increased levels of blood-borne, circulating factors that stimulate proliferation of HSCs. In particular IL-1β, binding the receptor (IL-1R1) expressed by HSCs and non-hematopoietic cells of the niches, induced activation of the BM and production of neutrophils and monocytes [85].

Together with IL1β, several other factors act directly on hematopoiesis during the inflammatory phase of the MI including granulocyte colony-stimulating factor (G-CSF), granulocyte–macrophage colony-stimulating factor (GM-CSF), CXCL12 and CCL2. These factors can modulate proliferation and differentiation of hematopoietic progenitors during the course of MI [62,92].

G-CSF induces maturation, survival, proliferation, and functional activation of granulocytes. Moreover, it also plays a crucial role in the mobilization of granulocytes and HSPCs from the BM into the peripheral circulation [62,93]. Indeed, G-CSF inhibits macrophage-mediated retention signals in the BM and enhances SNS-mediated progenitor release [82]. As mentioned before, the SNS has a fundamental role in homeostatic hematopoiesis. Additionally, the SNS is also involved in the regulation of the BM in case of MI. Noradrenaline increased in the BM of mice after MI. As consequence, Nes^+^ MSC, expressing β3AR, decrease the expression of Cxcl12 and Scf. The downregulation of those retention factors stimulates the release of HSCs into the blood [94]. Due to the capability to regulate different processes, the role of G-CSF in case of MI has been extensively studied, especially as possible pro-regenerative treatment. However, results from studies that evaluated the therapeutic potential of G-CSF are conflicting. Several studies showed that G-CSF treatment, administered before and after MI onset, may be beneficial inducing a better regeneration of cardiac tissue. On the other hand, more recent studies showed that G-CSF fuels inflammation in the cardiac tissue [95,96,97].

GM-CSF is produced by different cell types including leukocytes, fibroblasts and ECs. Like in the case of G-CSF, GM-CSF induces BM cell mobilization and was thus studied as pro-regenerative factor after MI. Alone or in combination with other cytokines, G-CSF can stimulate survival, proliferation and differentiation of different myeloid cells like neutrophils, eosinophils, monocytes and DCs. In particular GM-CSF, in combination with tumor necrosis factor-α (TNFα) and IL4, induces the differentiation of myeloid cells into DCs [93]. Although GM-CSF may confer cardiac reparative benefits when administered together with other growth factors, current evidence suggests that GM-CSF-induced cellular infiltration and molecular changes cause more damage than benefits in the post-infarction period. A study conducted by Anzai and colleagues showed that cardiac fibroblasts are the major source of GM-CSF in the first days after MI, in both mice and humans. GM-CSF is produced in the infarcted heart and promotes myelopoiesis in the BM by targeting a specific progenitor subset that gives rise to neutrophils and Ly6C^HI^ monocytes. Moreover, the same study showed the GM-CSF impairs wound healing through increased leukocyte accumulation after MI (primarily neutrophils and monocytes). Indeed, GM-CSF induces expression of immune cell-attracting chemokines such as Cxcl2, Ccl2, Ccl3, Ccl5, and Ccl7 [98].

HSCs express MCSF1R, the receptor for macrophage colony-stimulating factor (M-CSF), and react to M-CSF stimulation with differentiation toward monocytes in vitro. Indeed, M-CSF is a cytokine that regulates the differentiation, proliferation, and survival of monocytic progenitor cells [99,100]. In steady state mice, perivascular stromal cells and osteoblastic cells of the hematopoietic niches are the major sources of M-CSF [101]. The comparison of two transgenic mouse lines, Csf1r^−^/Csf1r^−^ and Csf1^op^/Csf1^op^, confirmed the fundamental role of M-CSF in the production and differentiation of monocytes. Both mouse lines showed a reduced number of circulating monocytes and a decreased number of tissue resident macrophages [102]. In case of MI, different animal models showed an increased expression of M-CSF in cardiac tissue [103]. Another study showed that the administration of recombinant M-CSF in mice with MI induces the egress of monocytes from the BM via modulating the CXCR4-CXCL12 axis [104]. Although we are still far from completely understanding the role and dynamics of M-CSF in hematopoiesis during MI, all data support the idea that M-CSF is involved in the modulation of hematopoiesis in response to the cardiac ischemia. Moreover, we cannot exclude the possibility that all mentioned growth factors act synergically with other molecules. Indeed, a study showed that G-CSF^−/−^, GM-CSF^−/−^ and M-CSF^−/−^ mice possess circulating monocytes, tissue macrophages and neutrophils, although the total number of myeloid cells was reduced when compared with control mice. This indicates that together with G-CSF, GM-CSF, and M-CSF, additional factors can stimulate myelopoiesis and act during acute inflammatory responses [105].

### 3.2. Development of Alternative Phenotypes of Myeloid Cells (Qualitative Alterations)

Several studies showed that pathological conditions such as infections, tumors or tissue injuries stimulate the production of myeloid cell with alternative phenotypes [25,106,107]. For instance, in a mouse model of cancer, distinct subpopulations of neutrophils have been identified. These different subpopulations can be distinguished according to their densities: “normal” and “low-density” neutrophils (LDNs). LDNs showed features associated with pro-tumor activity [108]. Another study identified a subtype of neutrophil with DC-like phenotype which express MHCII and CD11c. DC-like neutrophils differentiate from canonical neutrophils (with a rate <2%) in humans and mice under local and systemic inflammation [109]. Similar to neutrophils, monocytes with alternative phenotypes were identified. For instance, monocytes may develop a “neutrophil-like” phenotype (NeuMo) in response to LPS injection or monocytes may express Ceacam1 and Msr1 (SatM) in case of pulmonary fibrosis [110,111]. Further, a subtype of Ly6C^HI^ monocytes, expressing Sca-1 and MHCII, was found to be induced by interferon gamma (INFγ) [112]. Finally, Ym1^+^Ly6C^HI^ monocytes are released from the BM during the resolution phase of colitis [113] (Figure 3).

To our knowledge, there are no evidences regarding particular subsets of circulating monocytes or neutrophils triggered by MI, but future studies may fill this gap [90,107]. The general view is that myeloid cells acquire effector functions as a result of interactions with local signals/the local environment in the ischemic area of the heart. However, a recent study showed that the type I interferon (IFN) response to ischemic cardiac injury begins “remotely” already in the BM. Specifically, the authors showed that myeloid cells are characterized by the expression of IFN-stimulated genes after the MI, already at the level of the BM. This response occurs not only in monocytes or monocyte-derived macrophages, but also in neutrophils [114]. Another work identified the expansion of particular subsets of monocytes (CD14^+^ HLA-DR^−^) and immature neutrophils (CD16^+^ CD66b^+^CD10^−^) in the blood of patients with MI [115]. These two studies provide a clear confirmation that MI induces quantitative and qualitative changes in the myeloid repertoire.

### 3.3. Extramedullary Emergency Source of Myeloid Cells

Although the BM is the primary tissue responsible for the production of leukocytes, extramedullary hematopoiesis takes place in the spleen after the onset of the MI. As shown by Swirski and colleagues, the spleen is a reservoir of Ly6C^HI^ and Ly6C^LOW^ monocytes and, in case of splenectomy, leukocyte supply to the infarcted heart is reduced [116]. Within the first days after MI, a huge quantity of monocytes is released from the splenic subcapsular red pulp into the circulation. The mobilization of monocytes is driven by the interaction of angiotensin II with its receptors expressed by monocytes [117]. Consistently, the recent development of a high-density lipoprotein-derived nanotracer, which allows non-invasive in vivo tracking of myeloid cells, demonstrates that in case of permanent or transient LAD ligation, a significant quantity of myeloid cells is mobilized from the spleen [118]. As a consequence, the number of splenic Ly6C^HI^ monocytes decrease starting from day 1 post-MI and is restored within 4 days [119]. Splenic monocytes derive from maturation of HSPCs which egressed the BM and seeded the spleen prior. HSPCs are retained in the spleen by the presence of macrophages expressing vascular cell adhesion protein-1 (VCAM-1) [120]. Further, Grisanti and colleagues using chimeric mice with β2AR knockout reconstituted BM showed that the deletion of the β2-adrenoreceptor (β2AR) induces retention of leukocytes in the spleen due to an overexpression of VCAM-1. Moreover, the expression of both VCAM-1 and CCR2 appears to be regulated by β2AR signaling [121,122]. Further, these and others studies demonstrated the efficacy of β-blockers as treatment for MI [122,123,124]. Finally, the administration of bromodeoxyuridine (BrdU) showed a robust expansion of myeloid progenitor cells (MDPs) in the spleen after LAD ligation. Similar to the BM, IL-1β regulates the production of splenic monocytes after MI [125]. 

Splenic neutrophils take part in the response to bacteria, like in the case of *Streptococcus pneumoniae* infection, and it has been demonstrated that pancreatic carcinoma or sepsis causes extramedullary granulopoiesis [126,127]. Unlike monocytes, it is not entirely clear whether the spleen also functions a source of neutrophils in case of MI, so further studies are needed to clarify this aspect.

## 4. Mobilization and Recruitment of Circulating Myeloid Cells 

In healthy adult mice, the leukocyte count ranges from 2000 to 10,000 cells per microliter of blood. Neutrophils comprise 20% to 30% of the leukocyte count and are the most common granulocyte subset, while monocytes represent less then 2% of the total white blood cells [128]. Following MI, circulating myeloid cell counts peaked at day 2, mostly due to the expansion of neutrophils and Ly6C^HI^ monocytes [119]. The variation of the quantity of circulating leukocytes is the result of the regulation of two fundamental processes: (1) the release of the leukocytes from reservoir organs into the blood, and (2) the recruitment of blood leukocytes into the ischemic area. These two processes are regulated by a complex interplay of cytokines (such as chemokines, interleukins, TNFα and IFNγ), which regulate production, release, transportation, and uptake of leukocytes across involved tissues. The cytokines bind specific receptors expressed by leukocytes and other cell types, such as CMs and ECs. Chemokines act as BM-egression stimuli or chemoattractants. Similarly, inflammatory cytokines modulate leukocyte recruitment to the infarcted heart and the activation state of inflammatory cells [37,129,130].

### 4.1. Chemokines

Inflammatory chemokines are released from injured tissues. Their expression is induced by exposure to damage associated molecular patterns (DAMPs), reactive oxygen species (ROS) and C-peptides (derived from the complement cascade activation). Chemokines can be divided into four subgroups, however the CC and CXC families are best studied in case of MI [37,129].

#### 4.1.1. CC Chemokines

CC motif chemokine receptor 2 (CCR2), which binds C-C motif chemokine ligand 2 (CCL2, also termed monocyte chemoattractant protein-1, MCP-1), plays a fundamental role in the mobilization and recruitment of monocytes to the ischemic heart. Indeed, CCR2 is required for the release of Ly6C^HI^ monocytes from the BM into the blood [131]. A CCR2-silencing strategy, using lipid nanoparticle carrying short interfering RNA (siRNA), showed to reduce inflammatory monocytes levels in blood and heart after MI. Moreover, the administration of CCR2 siRNA induced the accumulation of monocytes in spleen and BM [132]. Monocytic expression of CCR2 is regulated by several signals. For instance, the β2-adrenergic receptor (β2AR) regulates CCR2 expression, thus the pharmacological and genetic block of the receptor decreased CCR2 expression and hence the quantity of monocytes/macrophages and neutrophils in the heart following MI [123]. A recent study showed that regulatory B cells reduced the expression of CCR2 on monocytes causing a decrease of monocytes mobilization from the BM and a decline in accumulation in the heart [133]. CCL2 can be produced by different cell types and its expression is induced by different inflammatory signals such as IL-1, IL-6, TNFα, IFNγ [134]. In the infarcted heart, CCL2 is predominantly expressed by ECs and infiltrating leukocytes. Experiments with CCL2 knockout mice or administration of anti-CCL2 antibodies showed that this factor has prominent effects on monocytes/macrophage recruitment and activation [135]. An alternative ligand of CCR2 is CCL7. B cell-derived CCL7 levels increase in the blood during the inflammatory phase after MI. CCL7 induces monocytes mobilization from the BM and their recruitment to the heart further stimulating the inflammatory reaction. Thus, CCL7 and B cell depletion showed to be beneficial for heart function after MI [136].

The binding of CCL5/RANTES to CCR1, CCR3, or CCR5 was shown to orchestrate the recruitment of monocytes and neutrophils to the infarct site. Montecucco and colleagues showed that CCL5 levels rise systemically and locally in case of MI. Anti-CCL5 antibody treatment significantly reduced neutrophil and monocyte infiltration on day 1 and 3 in cardiac ischemia/reperfusion (I/R) injury and chronic ischemia mouse models. Moreover, the plasma concentration of other chemokines (CXCL1, CXCL2 and CCL2) were also affected when CCL5 was depleted. More recently, the blockage of the heteromeric CCL5-CXCL4 showed to reduce the number of monocytes and neutrophils in I/R injury mice [137,138].

#### 4.1.2. CX Chemokines

As mention before, the CXCL12/CXCR4 axis has a pivotal role in retention/mobilization of HSCs in/from the BM. MI reduced CXCL12 expression in the BM through SNS activity and signaling [92,94]. Moreover, CXCL12/CXCR4 represents a retention signal that prevents neutrophil egress from the BM [139]. Interestingly, the expression of CXCR4 is induced in “aged” neutrophils. The increased expression of CXCR4 causes neutrophil homing to the BM, thus neutrophil clearance from the blood [140]. After MI, CXCL12 expression increases in the infarct area and acts as a chemoattractant signal for circulating BM-derived cells [141].

While the CXCL12-CXCR4 axis regulates the retention of leukocytes in the BM, the interaction of CXCL1, -2, -8 with CXCR1 and CXCR2 promotes mobilization of neutrophils from the BM into the blood [139]. Interestingly, CXCR2 expression follows a circadian rhythm and, in case of MI, neutrophils showed different BM egress relatively to the time-of-day at MI onset [40]. A recent study demonstrated that the CLEC4E-receptor regulates CXCR2 activity. Indeed, Clec4-/- mice showed a reduced chemotaxis and hence reduced recruitment of neutrophils to the ischemic area [142]. Another known ligand of CXCR2 is CXCL8 (or IL-8). Endothelial CXCL8 acts as a chemoattractant and consequently fuels recruitment and activation of neutrophils [37,130].

### 4.2. Inflammatory Cytokines: IL-1 Family, IL-6 and TNFa

The concentration of IL-1β, IL-6 and TNFα rises in patients with MI. These inflammatory cytokines act as immunomodulatory factors in autocrine, paracrine and endocrine ways. The common feature of these cytokines is the activation of the TF nuclear factor kappa-light-chain-enhancer of activated B cells (NF-κB), which promotes the expression of proinflammatory genes [130].

A growing body of evidence suggests a fundamental role for IL-1β in the pathogenesis of ASCVD (atherosclerotic cardiovascular disease) [143]. The secretion of active IL-1β is induced by the formation of the inflammasome. In particular, the NLRP3 inflammasome is a protein complex composed by (1) NLRP3, (2) the apoptosis-associated speck-like protein containing a caspase activation and recruitment domain (ASC) and (3) caspase-1. As result of the activation of the inflammasome, the inflammatory cytokines IL-1β and IL-18 are released into the extracellular space [144]. Several studies demonstrated that the assembly of the NLPR3 inflammasome takes place in the injured cardiac tissue (mainly in ECs and fibroblasts). Considering that inflammasome activation leads to the release of cytokines which magnify the inflammatory process, the deletion of its components or the pharmacological inhibition restored left ventricular function better after MI [85]. In NLPR3, ASC and caspase-1 deficient mice, IL-1β and IL-18 was reduced and consequently cardiac inflammation, myocardial fibrosis and cardiac dysfunction were markedly reduced in infarcted mice [145,146,147]. 

As mentioned before, IL-1β stimulates the production of myeloid cells and regulates leukocyte recruitment. Indeed, in case of administration of an anti-IL-1β antibody, fewer recruited myeloid cells have been detected in the ischemic area, on day 3 and 7 after LAD ligation [85]. In line with these data, IL-1R1^−/−^ mice showed reduced leukocytes infiltration and chemokine expressions [148,149]. Together with IL-1β and IL-18, IL-1α is part of the IL-1 family, although its maturation is caspase-1 independent. A recent work showed that IL-1α is predominantly expressed by CMs and cardiac neutrophils and that IL-1α induces endothelial expression of VCAM-1, a membrane protein involved in leukocyte recruitment. Experiments with Il1a^−/−^ mice showed a reduced number of macrophages and neutrophils in the ischemic area [150].

Similar to IL-1β, IL-6 takes part to the inflammatory phase after MI. In the classical signaling, IL-6 stimulates target cells by binding to IL-6R (CD126), which is associated with gp130 (CD130). On the other hand, the trans-signaling is initiated by the interaction between IL-6 and the soluble form of the receptor (sIL-6R) and resulting IL-6/sIL-6R complexes bind gp130 [151]. In both cases, gp130 dimerizes causing the activation of intracellular cascades. IL-6 has multiple roles in the inflammatory phase post-MI, moreover, it is also involved in the reparative phase. After infarction, IL-6 concentrations rise both in plasma and the cardiac tissue due to the production by infiltrated leukocytes (monocytes, macrophages and neutrophils), ECs and CMs. During the inflammatory process, IL-6 activates recruited neutrophils and stimulates the expression of intercellular adhesion molecule-1 (ICAM-1, adhesion molecule involved in leukocyte recruitment) by cardiac cells [152,153]. Using a rat model of MI, a recent study showed that IL-6 has a biphasic kinetic in the plasma: a first wave of IL-6 is caused by its release from cardiac cells of the ischemic area. In a second wave, IL-6 is liberated from invaded leukocytes. Moreover, George and colleagues demonstrated that the blockade of IL-6 trans-signaling reduces local release of CCL2 (but not in the plasma) and the quantity of recruited neutrophils and macrophages in the myocardium [154]. Similarly, the administration of IL-6R antagonist to mice with permanent coronary occlusion decreases neutrophil and macrophage infiltration and attenuates matrix metalloproteinase (MMP) activation leading to higher survival rate, improved left ventricular dilatation and contractile function compared to the control group [155].

TNFα has bivalent functions in case of MI: when it binds TNFR1, it leads to CM apoptosis and cardiotoxicity; when it is combined with TNFR2, it inhibits the inflammatory reaction and reduces CM apoptosis [156]. In steady state hearts, the concentration of TNF-α is mainly controlled by ECs and resident mast cells. In case of MI, preformed TNF-α is released within minutes from cardiac resident mast cells and macrophages [157,158]. Using TNF^−/−^ mice, Sun and colleagues showed that, during the inflammatory phase post-MI, TNF induces the recruitment of leukocytes and the expression of MMP9, IL-1β and IL-6 [159]. Moreover, an in vitro study showed that both IL-1β and TNFα increased the expression of ICAM-1 and VCAM-1 in CMs and fibroblasts [160].

In summary, the expression and the roles of inflammatory cytokines after the MI seems to be substantially interconnected. Their synergic actions induce the amplification of inflammation and regulate the recruitment of the myeloid cells to the ischemic area. For these reasons, several studies demonstrated that they may represent therapeutic targets [85,151,161].

## 5. Myeloid Cells in Healthy and Infarcted Hearts

An analysis of the cells that populate the heart reveals that leukocytes represent around 10% of the non-CM compartment. The majority of leukocytes are macrophages, while the count of lymphoid cells is minimal [162]. Moreover, also mast cells and DC have been identified in the heart, especially in cardiac valves [158,163]. Sc-RNAseq analysis of adult murine hearts allowed to identify the major types of cardiac leukocytes based on their gene expression profiles: granulocytes (Ccr1^+^, Csf3r^+^, S100a9^+^), lymphocytes (Ms4a1^+^, Cd3e^+^, Klrb1c^+^, Ncr1^+^), macrophages (Adgre1^+^, Fcgr1^+^) and dendritic cell (DC)-like cells (Cd209a^+^) [164]. Further studies showed high heterogeneity within the macrophage population and identified different subsets: CCR2^−^MHCII^HI^ (H2-eb1^+^), CCR2^+^ (Ccr2^+^), Timed4^+^ (Timd4^+^, Lyve1^+^ and Igf1^+^) and INF signaling-related macrophages (Irf7^+^, Isg20^+^ and Ifit1^+^) (Figure 5). In the same study, using a transgenic line with tamoxifen-inducible Cx3cr1^CreERT2–IRES–YFP^, the authors showed that in adult mice, CCR2 identified the subset of macrophages that originated from recruited circulating/BM-derived monocytes [165]. These data confirm previous results which showed that cardiac resident macrophages are mostly CCR2^−^ and derived minimally from circulating monocytes in physiological conditions. Resident macrophages arise from embryonic yolk-sac progenitors before birth and are self-maintaining, so independent from BM–derived monocytes [166,167]. The functions of resident macrophages include removal of apoptotic stromal cells and surveillance of the host tissue. Interestingly, among macrophages, Skelly and colleagues discovered a fifth subpopulation of macrophages with a hybrid molecular signature of macrophages and fibroblasts (Col1a1^+^, Pdgfra^+^, Tcf21^+^ and Fcgr1^+^, Cd14^+^, and Ptprc^+^). The presence of this subpopulation in the heart, named fibrocyte, has been further confirmed using the gene reporter mouse PDGFRa^GFP/+^. Moreover, it has been reported that fibrocytes could be involved in the organization of the cardiac matrix scaffold by interacting with fibroblasts [164,168] (Figure 4).

### 5.1. Cardiac Myeloid Cells during the Inflammatory Phase

With MI onset, the immunological landscape of the heart change drastically. Early after MI, resident cardiac cells (CMs, fibroblasts, macrophages, and mast cells) start to produce pro-inflammatory factors such as TNFα, IL-1, IL-6 and Ecs become activated triggering the recruitment of pro-inflammatory blood myeloid cells. Within a few hours, neutrophils start to populate the ischemic area. Activated neutrophils generate large amounts of ROS, which cause tissue injury by modifying amino acids, proteins, and lipids. Further, ROS stimulate the release of pro-inflammatory factors in the ischemic myocardium. Upon degranulation, neutrophils release a wide range of pre-synthesized granular proteins including myeloperoxidase (MPO), serine proteases, and MMPs. These enzymes can cause myocyte death and ECM degradation. Finally, neutrophils can secrete TNF-α, IL-1β, IL-8 CXCL1, 2, 3, and 8, which amplify local inflammation [32,33,170]. Activated neutrophils are also able to release neutrophil extracellular traps (NETs), which are composed of decondensated chromatin in association with elastase, MPO and cytoplasmic proteins. This process is called NETosis. Peptidylarginine deiminase 4 (PAD4) has a fundamental role in NETosis since it allows chromatin decondensation [171]. In infarcted PAD4^−/−^ mice, it was demonstrated that the absence of PAD4 afflicts the release of extracellular chromatin. Reduced NETosis had a cardioprotective effect during the first 24 h post-MI [172]. Sc-RNAseq analyses of cardiac neutrophils after MI highlight their heterogeneity. Indeed, between day 1 and 3 after MI, neutrophils (S100a8^+^, S100a9^+^, Retnlg^+^, Mmp9^+^, Cxcr2^+^) could be divided in several clusters. The majority of neutrophils expressed classical genes such as Lcn2, Camp, Retnlg and Mmp8. In addition to classical neutrophils, three other clusters have been identified: (1)Neutrophils expressing genes that are associated with NF-kB activation, including Nfkb1, Icam1, Il1a, Sod2, and Tnip1;(2)Neutrophils expressing genes linked with hypoxia-inducible factor 1α (HIF-1α) activation including Egln3, Hilpda, and Vegfa;(3)Neutrophils exhibiting an IFN response signature (Isg15^+^, Rsad2^+^, Ifit1^+^) [173,174]. It is not clear yet, whether the NETotic neutrophils belong to one of these clusters or have been excluded from the analysis since they are dying cells (Figure 5). A recent analysis of the transcriptome of neutrophils revealed that TFs like RELB, IRF5 and JUNB contribute to neutrophil survival and activation at the site of injury and promote phagocytosis, ROS production and NETosis. Finally, JUNB-deficient neutrophils showed reduced IL1β and ROS production leading to reduced infarct size [175].

Following MI, cardiac resident macrophages die and are replenished by blood monocyte-derived macrophages. Indeed, starting from the first day post-MI, BM- and spleen-derived CCR2^+^ Ly6C^HI^ monocytes invade the ischemic area [88,166,176]. Here, the recruited monocytes release proteolytic enzymes and MMPs and secrete proinflammatory cytokines contributing to further degradation of the ECM and amplification of inflammatory signaling. Finally, recruited monocytes give rise to pro-inflammatory macrophages. Monocyte-derived macrophages (Adgre1^+^, Itgam^+^, Fcgr1^+^, Ly6c2^+^, Ccr2^+^) release IL-1β, IL-6 and TNFα, amplify inflammation, and phagocytose dying cells [30,177]. A recent study combined sc-RNAseq data from three different reports [164,177,178] on cardiac non-CM cells after MI. Focusing on leukocytes, the authors showed that resident cardiac macrophages (Cx3cr1^+^, Lyve1+ and Mrc1^+^) and monocytes-derived macrophages (Ccr2^+^, Ccl2^+^, Ccl7^+^ and Ly6c2^+^) were greatly different in terms of gene expression, although previous results showed that CCR2^+^ macrophages have nearly identical transcriptional signatures to macrophages in steady-state mice [165,179]. Moreover, the integrative analysis revealed that monocytes-derived macrophages were characterized by the expression of genes related to exocytosis and myeloid cell activation. The analysis of the TFs that influence both populations of macrophages demonstrate that Stat1 and Irf7 are predominantly expressed by inflammatory macrophages, while Erg1 and Jund are specific regulons of resident macrophages [179]. Additionally, Farbehi and colleagues showed the presence of additional subpopulations of macrophages at day 3 post-MI, such as neutrophil-like macrophages (S100a9^+^ and Csf3r^+^) and IFN signaling related macrophages (MAC-ISG) corresponding to inflammatory macrophages described by King and colleagues [178,179] (Figure 5).

#### Amplification of the Inflammation: DAMPs

During the inflammatory phase post MI, the deregulation of oxygen supply induces apoptosis and necrosis of CMs. Both mechanisms of cell death lead to release of DAMPs, endogenous proteins, released into the extracellular environment [22,180]. S100 family members (S100A9 and S100A8), high mobility group box 1 (HMGB1), adenosine triphosphate (ATP), heat-shock proteins (HSPs), extracellular DNA and RNA are all types of DAMPs. In case of MI, DAMPs act on cardiac myeloid cells and bind to pattern recognition receptors (PRRs), which in turn triggers signaling cascades that activate TFs such as NF-κB, activator protein 1 (AP-1) and INF regulatory factor (IRF). Moreover, DAMPs can activate intracellular NOD-like receptors (NLRs) promoting the formation of the inflammasome [22,37,180,181,182].

HMGB1 is a ubiquitous nuclear binding protein that can regulate the structure and stability of chromosomes. Dying CMs release HMGB1, which can bind Toll-like receptor-2 and -4 (TLR2, TLR4) and receptors for advanced glycation end-products (RAGE) expressed by neutrophils and macrophages [183]. Rat experimental MI models showed that the blockade of HMGB1, by the administration of a neutralizing anti-HMGB1 antibody, decreased numbers of cardiac macrophages 3 days post-MI [184]. In an I/R injury mouse model, treatment with anti-HMGB1 antibody and/or Dnase I showed to reduce infarct size. However, the combined treatment did not produce additional reduction of the ischemic area when compared with the individual treatments [185].

During I/R injury, CMs die primarily via necrosis, which causes DNA release into the extracellular space and blood. The circulating DNA is recognized by TLR2 and TLR4 expressed by leukocytes [186]. Moreover, during the ischemia, ROS induce CM release of mitochondria DNA (mtDNA), which can be recognized by TLR9, RAGE and NLPR3 inflammasome. Significant correlations between elevated plasma level of mtDNA and expression levels TNF, IL-6 and IL-8 were observed in myocardial tissue following I/R injury [187].

The S100 family is a group of cytoplasmatic proteins containing calcium-binding motifs, which can be released into the extracellular space. S100A8 and S100A9, which can form heterodimers, have been studied intensively due to their pathophysiological role in MI [181]. Cardiac neutrophils are the major source of S100A8/A9, which modulate leukocyte recruitment and prime the inflammasome through binding TLR4 and RAGE [188]. The administration of ABR-238901 (S100A9 blocker) during the inflammatory phase of MI decreased numbers of macrophages and neutrophils promoting an anti-inflammatory environment in the heart [189].

### 5.2. Cardiac Myeloid Cells in the Resolution of Ischemic Injury

Starting from day 4 post-MI, the reparative phase takes place and myeloid cells that populate the ischemic area play a fundamental role in its regulation. Neutrophil numbers decrease, while macrophages adopt an anti-inflammatory and pro-regenerative phenotype [23] (Figure 5). Neutrophils in particular contribute to the polarization of macrophages toward a reparative phenotype. Indeed, induction of MER Proto-Oncogene Tyrosine Kinase (MerTK) expression and increased efferocytosis capacity of macrophages is mediated via neutrophil secreted neutrophil-gelatinase associated lipocalin (NGAL) [190]. In addition, phagocytosis of apoptotic cells and concomitant activation of Smad3 decreases production of pro-inflammatory cytokines and induces the production of anti-inflammatory and pro-fibrotic cytokines, such as IL-10, TGFβ and angiogenic factors like VEGF [191]. Moreover, IL-4 stimulates the anti-inflammatory phenotypic shift of macrophages, which acquire the expression of Ym1 (Chil3) and Arg1 [192] (Figure 5). Thus, the resolution of inflammation triggers the reparative phase which is characterized by activation of myo-/fibroblasts and formation of new vessels resulting in the formation of a scar after several weeks. Activated and highly proliferative fibroblasts peak within 4 days after MI. By day 4 to 7, these cells differentiate into myofibroblasts which secrete abundant extracellular matrix proteins and express smooth muscle α-actin to structurally support the necrotic area. By day 7 to 10, myofibroblasts lose the proliferative ability leading to maturation of the cardiac scar [36]. Two weeks post-MI, the quantity of monocytes and macrophages returns to baseline, however macrophages persist in the remote area for months after the infarction [86].

The analysis of the immunological landscape shows how drastically the cardiac myeloid cell composition changes in response to MI. It is now clear that the heterogeneity of leukocytes cannot be reduced to a simple change in polarization (M1 vs. M2). Indeed, different myeloid cell subtypes are involved in cardiac healing after MI. Further research is needed to characterize the functional aspects of this immunological heterogeneity to ultimately identify novel therapeutic targets to reduce MI-related morbidity and mortality.

## 6. Conclusions and Future Perspective

MI provokes a dramatic local and systemic inflammatory response inducing alterations not only in the heart itself, but also in remote organs. These organs are interconnected by complex inflammatory networks consisting of extracellular factors like cytokines, chemokine and danger signals which orchestrate leukocyte trafficking [22,37,45]. Scientists have to comprehend these immunodynamics to identify key regulators of the inflammatory process to beneficially modulate resolution of cardiac injury. Emerging therapeutic strategies aim to inhibit innate immunity targeting cytokines such IL-1β or IL-6 in CVDs [161,193,194]. In that light, the CANTOS (Canakinumab Anti-Inflammatory Thrombosis Outcomes Study) trial demonstrated that targeting IL-1β significantly lowers major cardiovascular events in 10,061 patients with previous myocardial infarction and elevated high-sensitivity C-reactive protein levels (hsCRP) [193]. Moreover, several trials and studies showed the efficacy of colchicine (which targets microtubules and inflammasome assembly) in patients and animal models of CVDs [195]. In particular, the Colchicine Cardiovascular Outcomes Trial (COLCOT) demonstrated that colchicine led to a significantly lower risk of ischemic cardiovascular events in patients with prior MI [196,197]. Additionally, in the Low-Dose Colchicine (LoDoCo)-2 trial, patients with chronic coronary syndrome treated with colchicine developed less cardiovascular events, such as cardiovascular death, spontaneous myocardial infarction, ischemic stroke, or ischemia-driven coronary revascularization [198]. To better understand the biological basis of these beneficial effects we recently explored colchicine’s actions in mouse models of atherosclerosis and MI [199]. We found that colchicine deactivated circulating neutrophils and monocytes and hence lowered recruitment of these cells into atherosclerotic plaques and the infarct area. Although the mentioned studies provided promising data in favor of colchicine as a treatment option for CVD, some aspects have to be further clarified, such as the balance between benefits and risks of immunomodulation [200]. RESCUE (IL-6 Inhibition with Ziltivekimab in Patients at High Atherosclerotic Risk) and ZEUS (Zlitivekimab Cardiovascular Outcomes Study) are two trials aiming to test the effect of ziltivekimab (a human monoclonal antibody directed against IL-6) in patients with high cardiovascular risk. Both trials compared the effect of ziltivekimab to placebo in patients with chronic kidney disease and elevated hsCRP to test whether depleting circulating IL-6 reduces cardiovascular events [201,202]. A recent study analyzed the effect of different doses of β-blockers on survival in case of MI. Patients treated with >12.5% to 25% of the targeted dose showed enhanced survival compared to no β-blocker therapy and other β-blocker doses [203]. β-blockers have been shown to impair recruitment of monocytes and macrophages to the ischemic area and to affect neutrophil activity [121,122,123,124,204,205]. All these studies demonstrate the pivotal role of inflammation and immunity in initiation and progression of CVDs. However, it is fundamental to take the balance between detrimental and reparative effects of immune cells after cardiac injury into consideration. Indeed, as a possible alternative to the inhibition of inflammatory signaling, Leone and colleagues showed in a 10-year follow-up study that STEMI patients treated with G-CSF displayed decreased prevalence of adverse left ventricle remodeling and rate of heart failure with improvement in quality of life. These results are in line with the previous RIGENERA study performed by the same group where they demonstrated that patients treated with G-CSF (10 µg/kg/day for 5 days) had higher left ventricle ejection fractions and lower left ventricular end-diastolic volumes compared to patients treated with conventional therapy five months after MI [206,207].

In summary, recent therapeutic advances greatly helped to reduce the global burden of CVDs. However, novel insights are required to clarify yet unknown features of the pathophysiology of MI and to improve the efficacy of interventional treatment. We illustrated that the immune response to ischemic cardiac injury starts in the bone marrow and spleen—organs that are located far away from the site of tissue injury—with emergency myelopoiesis. This systemic reaction is controlled by a complex network of extracellular signals which include cytokines, chemokines, DAMPs, and growth factors. Extracellular signaling impacts not only immunodynamics, but also the heterogeneity of leukocytes in different remotely located compartments and in the ischemic area. Indeed, the recent development of cell lineage tracing techniques and multiomics analysis led to a better understanding of the hematopoietic process and leukocyte heterogeneity during the course of the MI [46,164]. Thus, it is fundamental to give these findings a meaning. In other words, further studies have to clarify whether different subpopulations of myeloid cells have different roles during MI or merely represent a phenotypic shift without a particular function. Moreover, the comprehension of the signaling that shapes the cellular heterogeneity will be essential in finding new therapeutic strategies of targeting inflammation in CDVs.

The discussed pre-clinical and clinical trials greatly advanced the scientific field of cardiovascular inflammation and represent a huge step forward toward a better com-prehension of the inflammatory networks at work after MI. However, these studies do not always fully consider the heterogeneity of immune cells in hematopoietic organs and at the site of injury. New technologies, such as cell tracking and Sc-RNAseq, will greatly aid to identify novel druggable targets in MI treatment. Thus, future anti-inflammatory studies will need to carefully take into account: (1) the exact target leukocyte subpopulation, (2) when to initiate and (3) when to stop treatment after MI.

## Figures and Tables

**Figure 1 ijms-23-15814-f001:**
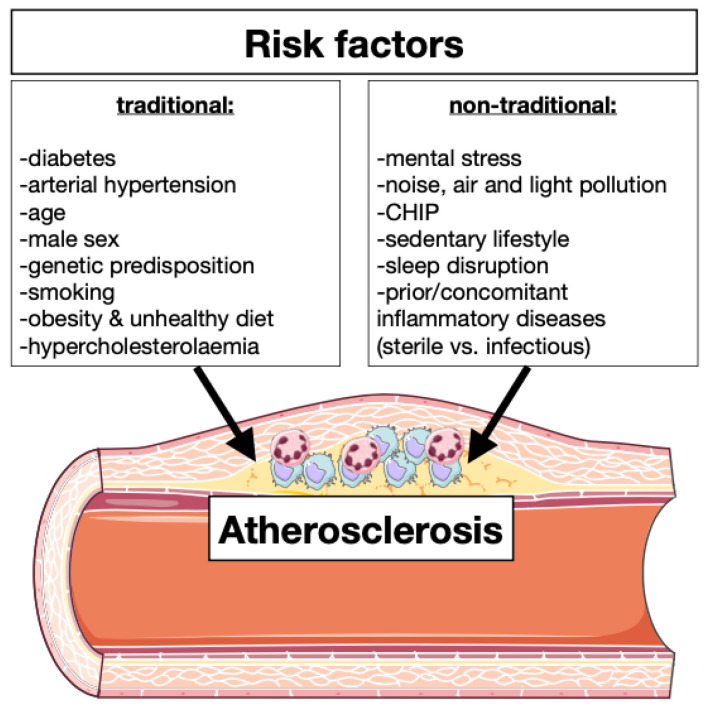
Traditional and non-traditional atherosclerosis risk factors. There are several risk factors for initiation and progression of atherosclerosis which can be divided in two different classes: traditional and non-traditional. Factors such as diabetes, hypertension, age, gender, genetic predisposition, smoking, obesity, unhealthy diet, and hypercholesterolemia are well studied and are there-fore classified as traditional risk factors for atherosclerosis. Mental stress, environmental elements (noise, air quality and light), clonal hematopoiesis of indeterminate potential (CHIP), sedentary lifestyle, sleep disruption and prior or concomitant inflammatory diseases have recently emerged as risk factors of atherosclerosis and are hence called non-traditional risk factors.

**Figure 2 ijms-23-15814-f002:**
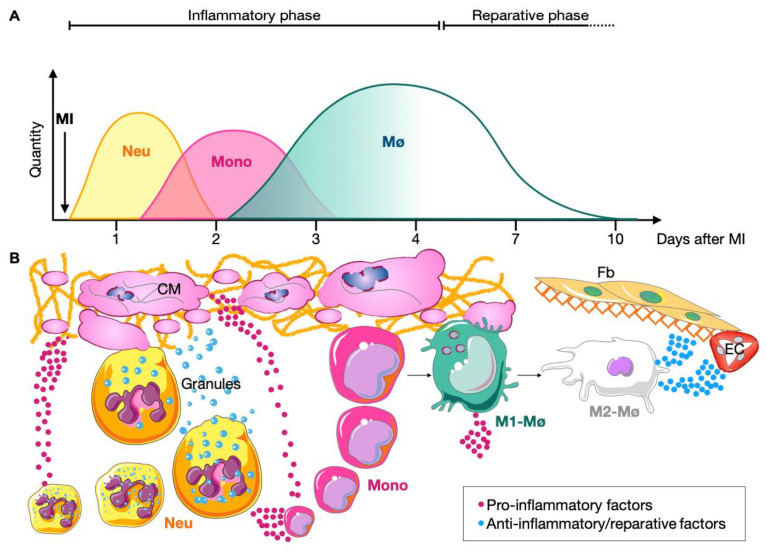
“Emergency crew” in the ischemic area. (**A**) Representation of the quantity of recruited neutrophils (Neu), monocytes (Mono) and macrophages (Mø) in the ischemic area during the inflammatory (approximately 3–4 days post-MI) and reparative (approximately from day 4 to 10 post-MI) phases. (**B**) Representation of the interactions between immune cells and the ischemic tissue during MI progression. With the onset of the infarct, cardiomyocytes (CM) die and release pro-inflammatory factors (magenta dots), which recall immune cells to the site of injury. Recruited neutrophils act as phagocytes removing cell debris, extracellular matrix and dying cells. Moreover, they release granules containing proteases and stimulate the recruitment of blood monocytes by releasing extracellular factors (magenta dots). Next, monocytes invade the ischemic area and give rise to inflammatory macrophages (M1-Mø), which phagocytose cellular debris, extracellular matrix and release pro-inflammatory factors (magenta dots). During the reparative phase, recruited macrophages change the polarization from pro-inflammatory to anti-inflammatory/reparative (M2-Mø). M2-Mø release pro-fibrotic and anti-inflammatory factors (blues dots) inducing the revascularization and the deposition of extracellular matrix allowing the repopulation of the ischemic area by endothelial cells (EC) and myofibroblasts (Fb).

**Figure 3 ijms-23-15814-f003:**
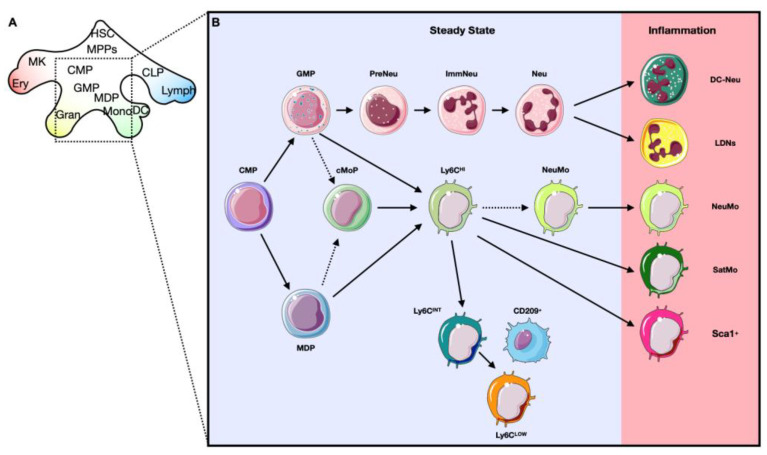
General development of hematopoietic cells and heterogeneity of neutrophils and monocytes in the steady state and in inflammatory conditions. (**A**) Representation of the hematopoietic process. The generation of mature blood cells starts with hematopoietic stem cells (HSCs) which give rise to the multipotent precursors (MMPs). From here: (1) on the erythroid branch, maturation of megakaryocytes (MK) and erythrocytes (Ery) takes place; (2) on the lymphoid branch, common lymphocyte precursors (CLPs) mature into the different type of lymphocytes (Lymph); (3) on the myeloid branch, granulocytes (Gran) derive from granulocyte macrophage precursors (GMPs), which originate from common myeloid progenitor (CMPs). Similarly, monocyte-macrophage/dendritic cell precursors (MDPs), which also derive from CMPs, can give rise to monocytes (Mono) and dendritic cells (DC). Both GMPs and CMPs can mature into Mono. (**B**) Representation of the maturation of neutrophils (Neu), monocytes and alternative leukocytes in case of inflammation. The development of neutrophils and monocytes starts with CMPs. The Neu maturation includes different steps: GMPs give rise to proliferative committed progenitors (Pre-Neu), which then develop into immature stage (ImmNeu) and later into the mature Neu. In a pathological state, Neu can acquire different phenotypes: low-density neutrophils (LDNs) or DC-like phenotype (DC-Neu). Murine monocytes can be divided into different classes based on the expression of Ly6C: classical (Ly6C^HI^), intermediate (Ly6C^INT^) and non-classical (Ly6C^LOW^). Among the Ly6C^INT^, some monocytes express MHCII and CD209. Ly6C^HI^ monocytes rise directly from GMPs and MDPs or from a common monocyte precursor (cMoPs). In case of inflammation, different subtypes of monocytes have been identified: monocytes expressing Ceacam1 and Msr1 (SatM), monocytes expressing Sca-1 and MHCII (Sca-1^+^) and neutrophil-like monocytes (NeuMo). It is not entirely clear whether Neu-Mo develop only in case of inflammation or in homeostatic condition as well (dashed arrow).

**Figure 4 ijms-23-15814-f004:**
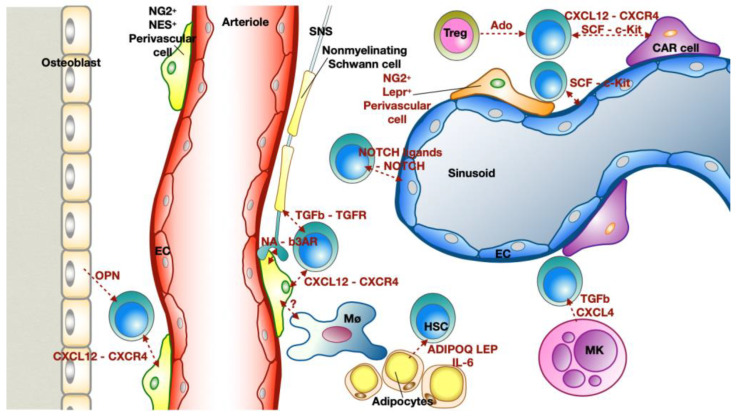
Bone marrow hematopoietic niche in steady state adult mice. Hematopoietic niches are located in the perivascular space and are associated with arterioles (red) or sinusoidal (blue) blood vessels of the bone marrow. The niches are composed of hematopoietic stem cells (HSCs), sinusoidal or arteriolar endothelial cells (ECs), perivascular mesenchymal cells (NG2^+^ NES^+^ perivascular cells, NG2^+^Lepr^+^ perivascular cells and CAR cells), fibers of the sympathetic nervous system (SNS), non-myelinating Schwann cells, adipocytes, osteoblasts and leukocytes. The hematopoietic components of the niche include: megakaryocytes (MK), macrophages (Mø) and regulatory T cells (Treg). The key signals for the maintenance of HSCs are the stem cell factor (SCF), which binds to c-Kit, and the CXC-chemokine ligand 12 (CXCL12), which binds to CXC-chemokine receptor 4 (CXCR4). The main sources of SCF and CXCL12 are ECs, perivascular CAR cells and NG2^+^ NES^+^ perivascular cells. Moreover, ECs regulate HSC self-renewal through NOTCH signaling. SNS secrete noradrenaline (NA), which down-regulates the expression of Cxcl12 in perivascular cells, binding the β3-adrenergic receptor (β3AR). Mø interact with the perivasculature in yet unknown ways inducing the retention of HSCs through antagonizing the mobilization signals of the SNS. Non-myelinating Schwann cells release transforming growth factor-β (TGFb), which binds to its receptor expressed by HSCs. BM adipocytes affect HSCs by secreting adiponectin (ADIPOQ), leptin (LEP) and interleukine-6 (IL-6). Osteoblasts are a major source of osteopontin (Opn), which is a negative regulator of HSC proliferation. MK are in close proximity to HSCs and regulate HSC quiescence by producing CXCL4 and TGFb. Treg are not directly in contact with HSCs, they regulate the maintenance of HSC by release of adenosine (Ado).

**Figure 5 ijms-23-15814-f005:**
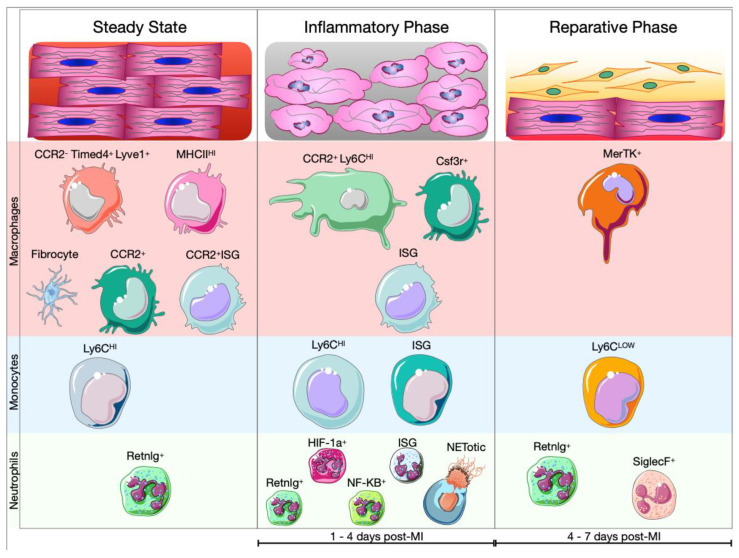
Cardiac macrophages, monocytes and neutrophils during the course of myocardial infarction. In the steady state, the heart is populated by different myeloid cells. Among them, macro-phages represent the majority. Different macrophage phenotypes have been identified: CCR2^−^ macrophages divided in (1) MHCIIHI and (2) Timd4^+^/Lyve1^+^, (3) CCR2^+^ macrophages and (4) interferon signaling related macrophages (CCR2^+^ ISG). A fifth subpopulation of macrophages is termed (5) fibrocytes since they express fibroblast-related genes. Additionally, the healthy heart is populated by a minimal number of monocytes (Ly6C^HI^) and neutrophils (Retnlg^+^). With the onset of the MI, the immunological landscape of the heart changes drastically and myeloid cells are recruited to the site of injury to remove cellular debris and extracellular matrix. At the beginning of the inflammatory phase (approximately from day 1 to 4 after MI), neutrophils invade the ischemic area. Besides (1) classical neutrophils (Retnlg^+^), four other neutrophil subtypes have been identified: (2) neutrophils characterized by genes associated with NF-kB activation (NF-kB^+^), (3) neutrophils expressing genes linked to HIF-1α (HIF-1a^+^), (4) neutrophils identified by IFN response signature (ISG) and (5) NETotic neutrophils. After MI, cardiac resident macrophages are replaced by monocyte-derived macrophages (CCR2^+^ Ly6C^HI^). Moreover, recent sc-RNAseq analyses highlighted the presence of minor populations of macrophages, such as neutrophil-like macrophages (Csf3r^+^) and INF signaling related macrophages (ISG). Similarly, recruited monocytes showed two main phenotypes: classical monocytes (Ly6C^HI^) and interferon signaling related monocytes (ISG). During the reparative phase (approximately between day 3 and 7 after MI), the quantity of neutrophil starts to decline. They can be divided into two main subpopulations: classical neutrophils (Retnlg^+^) and SiglecF^+^ neutrophils. Moreover, non-classical monocytes (Ly6C^LOW^) start to populate the injured tissue, meanwhile, inflammatory macrophages change their phenotype and become anti-inflammatory macrophages (MerTK^+^) [169]. Anti-inflammatory macrophages promote the activation of fibroblast and the revascularization of the ischemic area facilitating the formation of a scar.

**Table 1 ijms-23-15814-t001:** Traditional and non-traditional atherosclerosis risk factors which are linked to immunity.

Traditional Risk Factors	Effects on Immune System
Diabetes	Hyperglycemia leads to enhanced myelopoiesis [9]
Arterial hypertension	Increased quantity of circulating leukocytes and cytokine concentration [10]
Age	Clonal hematopoiesis, enhanced inflammation and endothelial disfunction [11]
Male sex	Testosterone induced inflammation and sexual immune dimorphism [12]
Genetic predisposition	Immunity modulation and leukocytes production [7]
Smoking	Leukocytes production and inflammasome activation [13]
Obesity and unhealthy diet	Adipose tissue macrophages induce leukocyte production [14]
Hypercholesterolemia	Production of foam cells, activates inflammasomes, enhanced immune signaling and oxidative stress [15]
**Non-Traditional Risk Factors**	
Mental stress	Leukocytes expansion and infiltration [5,8]
Noise, air, and light pollution	Oxidative stress, enhanced inflammation and endothelial disfunction [16,17]
CHIP	Increased of circulating leukocytes [11,18]
Sedentary life	Lower level of myokine and possible regulation of immunity [11,19]
Sleep disruption	Chronic systemic low-grade inflammation [20]
Prior/concomitant inflammatory disease	Modulation of leukocyte production, phenotype and activation [21]

## Data Availability

Not applicable.

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
