# Peer review of "Quo Vadis? Immunodynamics of Myeloid Cells after Myocardial Infarction"

_ijms, 2022, doi:10.3390/ijms232415814_

Round 1

Reviewer 1 Report

The authors present a contemporary review of the immunomodulatory aspect of myocardial infarction and focus on the leukocyte supply chain and dynamics of myeloid cells after AMI has occurred.

I think that the present review is overall well-written and contains relevant bibliographic data and it is rich with pertinent illustrations. It is also comprehensive enough.

However, in order to improve certain portions of the manuscript I would advise authors following changes to be implemented, as specifically outlined below:

1. I would suggest authors elaborate slightly more and expand a portion of the text regarding the mechanisms and immunomodulatory implications of ischemia-reperfusion injury.

2. Regarding the G-CSF treatment, authors should acknowledge that the longest available follow-up of such treatment (10 years) in a small cohort of patients with large anterior MI showed to be feasible, safe, and associated with the lower extent of LV remodeling and higher quality of life (please cite Leone AM et al. J Clin Med. 2020;9(4):1214.). Also, it would be great to provide a short theoretical framework on G-CSF use in CV diseases (please consider inserting D'amario et al. Pharmacol Res. 2018; 127:67-76. for inclusion). These are all pertinent references.

3. Regarding the colchicine, the authors acknowledge the COLCOT trial in the references (by Tardif et al.). However, this should be updated also with the relevant LoDoCo-2 trial by Nidorf et al. (NEJM. 2020). This was tested successfully in the chronic coronary syndrome population. However, it should be stated that the benefits and risks of immunomodulation by colchicine in CV disease are not yet fully understood (Galli et al. 2021. Eur Heart J Cardiovasc Pharmacother and D'amario et al. Clin Res Cardiol. 2021). These are all important implications to be briefly integrated and discussed.

4. More aspects of NLPR3 inflammasome should be emphasized in the text as activation of this system is very important in the pathogenesis of many CV conditions. This part is only briefly mentioned in the current text.

5. Regarding the ziltivekimab, it should be correctly added that it is tested in patients with high cardiovascular risk and renal dysfunction - the point of this drug is to show a reduction in events among high-risk CV patients and renal insufficiency who have demonstrated evidence of an active low-grade systemic inflammation.

Author Response

Reviewer 1

The authors present a contemporary review of the immunomodulatory aspect of myocardial infarction and focus on the leukocyte supply chain and dynamics of myeloid cells after AMI has occurred.

I think that the present review is overall well-written and contains relevant bibliographic data and it is rich with pertinent illustrations. It is also comprehensive enough.

However, in order to improve certain portions of the manuscript I would advise authors following changes to be implemented, as specifically outlined below:

  1. I would suggest authors elaborate slightly more and expand a portion of the text regarding the mechanisms and immunomodulatory implications of ischemia-reperfusion injury.

We thank the reviewer for the comment. We added the following paragraph (highlighted in the manuscript) where we illustrate the differences between permanent and temporary LAD ligation.

Page 5, lines: 120-140

“The development of murine models of MI helped to appreciate several pathological and immunological aspects of cardiac injury that were not known before. Permanent ligation of the left anterior descending (LAD) coronary artery or its temporary occlusion causing ischemia and reperfusion injury (IRI) represent the two main animal models to study MI. From a cellular point of view, these two methods differ in the type of injury. In case of permanent ligation, extreme and prolonged hypoxia induces blockage of aerobic metabolism leading to activation of apoptotic pathways in cardiomyocytes (CM). In case of IRI, the temporary hypoxia - followed by reperfusion - induces production of free radicals by CM and variation of their cellular pH which causes mitochondria swelling and necrotic cell death [37,38]. Comparing these two models, permanent ligation causes higher mortality, bigger infarct sizes and a more prominent drop of left ventricular metabolic volume, as revealed by PET imaging. Moreover, different studies showed that the time of day (circadian control) of the ligation affects the progression of the injury in case of permanent ligation, but not in IRI [39–41]. These distinctive aspects of these two surgical techniques also impact the immune response that is triggered by the injury. Indeed, the comparison of these two LAD ligation models showed that the quantity of accumulated leukocytes in the infarct area is increased in case of permanent occlusion, while the number of circulating monocytes did not differ between the two surgical strategies [42]. Similarly, another study showed that reperfusion reduced the total number of leukocytes recruited to the infarct area and impacted the temporal course of innate and adaptive immune responses. In particular, reperfusion shifts the peak of the innate immune response towards earlier time points and also dampens the adaptive immune response [43].”

  1. Regarding the G-CSF treatment, authors should acknowledge that the longest available follow-up of such treatment (10 years) in a small cohort of patients with large anterior MI showed to be feasible, safe, and associated with the lower extent of LV remodeling and higher quality of life (please cite Leone AM et al. J Clin Med. 2020;9(4):1214.). Also, it would be great to provide a short theoretical framework on G-CSF use in CV diseases (please consider inserting D'amario et al. Pharmacol Res. 2018; 127:67-76. for inclusion). These are all pertinent references.

We thank the reviewer for the recommendations. Regarding the mentioned papers, D'amario et al. Pharmacol Res. 2018; 127:67-76 was already cited with reference number 76 (now 96). As suggested, we added the following paragraph regarding the studies of Leone and colleagues (highlighted in the manuscript).

Pages 19-20, lines: 835-843

“Indeed, as a possible alternative to the inhibition of inflammatory signaling, Leone and colleagues showed in a 10-year follow-up study that STEMI patients treated with G-CSF displayed decreased prevalence of adverse left ventricle remodeling and rate of heart failure with improvement in quality of life. These results are in line with the previous RIGENERA study performed by the same group where they demonstrated that patients treated with G-CSF (10 µg/kg/day for 5 days) had higher left ventricle ejection fractions and lower left ventricular end-diastolic volumes compared to patients treated with conventional therapy five months after MI [207,208].”

  1. Regarding the colchicine, the authors acknowledge the COLCOT trial in the references (by Tardif et al.). However, this should be updated also with the relevant LoDoCo-2 trial by Nidorf et al. (NEJM. 2020). This was tested successfully in the chronic coronary syndrome population. However, it should be stated that the benefits and risks of immunomodulation by colchicine in CV disease are not yet fully understood (Galli et al. 2021. Eur Heart J Cardiovasc Pharmacother and D'amario et al. Clin Res Cardiol. 2021). These are all important implications to be briefly integrated and discussed.

We appreciate the comment of the reviewer. Accordingly, we added and discussed the mentioned references (highlighted in the manuscript).

Page 19, lines: 810-822

“In particular, the Colchicine Cardiovascular Outcomes Trial (COLCOT) demonstrated that colchicine led to a significantly lower risk of ischemic cardiovascular events in patients with prior MI [197,198]. Additionally, in the Low-Dose Colchicine (LoDoCo)-2 trial, pa-tients with chronic coronary syndrome treated with colchicine developed less cardio-vascular events, such as cardiovascular death, spontaneous myocardial infarction, is-chemic stroke, or ischemia-driven coronary revascularization [199]. To better understand the biological basis of these beneficial effects we recently explored colchicine’s actions in mouse models of atherosclerosis and MI [200]. We found that colchicine deactivated circulating neutrophils and monocytes and hence lowered recruitment of these cells into atherosclerotic plaques and the infarct area. Although the mentioned studies provided promising data in favor of colchicine as a treatment option for CVD, some aspects have to be further clarified, such as the balance between benefits and risks of immunomodulation [201].”

  1. More aspects of NLPR3 inflammasome should be emphasized in the text as activation of this system is very important in the pathogenesis of many CV conditions. This part is only briefly mentioned in the current text.

Based on the reviewer suggestion, we expanded this part including some studies on inflammasome activation and inhibition in murine model of MI (highlighted text).

Page 14, lines: 574-581

“Several studies demonstrated that the assembly of the NLPR3 inflammasome takes place in the injured cardiac tissue (mainly in ECs and fibroblasts). Considering that inflam-masome activation leads to the release of cytokines which magnify the inflammatory process, the deletion of its components or the pharmacological inhibition restored left ventricular function better after MI [85]. In NLPR3, ASC and caspase-1 deficient mice, IL-1β and IL-18 was reduced and consequently cardiac inflammation, myocardial fibrosis and cardiac dysfunction were markedly reduced in infarcted mice [145–147].”

  1. Regarding the ziltivekimab, it should be correctly added that it is tested in patients with high cardiovascular risk and renal dysfunction - the point of this drug is to show a reduction in events among high-risk CV patients and renal insufficiency who have demonstrated evidence of an active low-grade systemic inflammation.

We thank the reviewer for the specification. To clarify this point, we added the following sentence (highlighted text).

Page 18, lines: 826-828

“Both trials compared the effect of ziltivekimab to placebo in patients with chronic kidney disease and elevated hsCRP to test whether depleting circulating IL-6 reduces cardio-vascular events [202,203].”

Reviewer 2 Report

Dear Authors,

Thank you very much for the review article entitled "Quo Vadis? Immunodynamics Of Myeloid Cells After Myocardial Infarction".

Manuscript is well-structured, detailed, and comprehensive, however, I have several suggestions and comments which could improve the readability and attractivity of the article.

1. There is a lack of visual information related to the topic. I kindly recommend adding several relevant images, schemes, plots, diagrams, etc.  from the cited articles. It helps to drastically increase the visual attractivity of the article.

2. P. 2, Figure 1
I kindly recommend adding a specific table similar to Table 1 presented in the following study: Nahrendorf, M. Myeloid cell contributions to cardiovascular health and disease. Nat Med 24, 711–720 (2018). https://doi.org/10.1038/s41591-018-0064-0.

Moreover, lipoproteins and its effect on atherosclerosis risk could be described (LDL, HDL).

3. P. 3, line 64
I suppose that instead of "life spam" the Authors mean "life span".

4. P. 6, line 232, and P. 7, line 261
Please, clarify the spelling: NOTCH or Notch.

5. P. 15, subsection 5.1, line 686
I kindly recommend to move each cluster of neutrophils on the individual lines:
1)
2)
3)

6. P. 17, Subsection 6.1
This subsection is more like discussion, than conclusion, and I kindly recommend to displace this information into the main text.

7. P. 18, Subsection 6.2
As mentioned above, I recommend changing this subsection into an individual section entitled "Conclusion and future perspectives". Please, highlight the incompleteness of knowledge and the further research needed to carry out.

Based on the aforementioned, I suggest minor revisions prior to the acceptance of this review.

Author Response

Reviewer 2

Dear Authors,

Thank you very much for the review article entitled "Quo Vadis? Immunodynamics Of Myeloid Cells After Myocardial Infarction".

Manuscript is well-structured, detailed, and comprehensive, however, I have several suggestions and comments which could improve the readability and attractivity of the article.

  1. There is a lack of visual information related to the topic. I kindly recommend adding several relevant images, schemes, plots, diagrams, etc. from the cited articles. It helps to drastically increase the visual attractivity of the article.

We thank the reviewer for the suggestion. We included the following visual abstract to the manuscript.

  1. P. 2, Figure 1

I kindly recommend adding a specific table similar to Table 1 presented in the following study: Nahrendorf, M. Myeloid cell contributions to cardiovascular health and disease. Nat Med 24, 711–720 (2018). https://doi.org/10.1038/s41591-018-0064-0.

Moreover, lipoproteins and its effect on atherosclerosis risk could be described (LDL, HDL).

We appreciate the suggestion and added Table 1 “Traditional and non-traditional atherosclerosis risk factors which are linked to immunity” (page 2, line: 38).

Regarding the association of atherosclerotic plaque development and levels of lipoproteins, we included the risk factor “hypercholesterolemia” in the table. Moreover, we refer to “formation of foam cells”, which represent lipid-laden macrophages.

  1. P. 3, line 64

I suppose that instead of "life spam" the Authors mean "life span".

We apologize for the oversight and corrected the mistake.

  1. P. 6, line 232, and P. 7, line 261

Please, clarify the spelling: NOTCH or Notch.

We apologize for the mistake. We meant “NOTCH signaling” (Page 9, line 281).

  1. P. 15, subsection 5.1, line 686

I kindly recommend to move each cluster of neutrophils on the individual lines:

1)

2)

3)

We thank the reviewer for the suggestion. We modified the text accordingly (highlighted text).

Page 17, lines: 699-705

“In addition to classical neutrophils, three other clusters have been identified:

1) neutrophils expressing genes that are associated with NF-kB activation including Nfkb1, Icam1, Il1a, Sod2, and Tnip1;

2) neutrophils expressing genes linked with hypoxia­inducible factor 1a (HIF-1a) activation including Egln3, Hilpda, and Vegfa;

3) neutrophils exhibiting an IFN response signature (Isg15+, Rsad2+, Ifit1+) [160,161].”

  1. P. 17, Subsection 6.1

This subsection is more like discussion, than conclusion, and I kindly recommend to displace this information into the main text.

We thank the reviewer for the comment. We changed the text accordingly and merged paragraphs 6 and 7 (conclusion and final remarks) in one unique paragraph 6 entitled “Conclusion and future perspective”. Page 19, line: 797 (highlighted text).

  1. P. 18, Subsection 6.2

As mentioned above, I recommend changing this subsection into an individual section entitled "Conclusion and future perspectives". Please, highlight the incompleteness of knowledge and the further research needed to carry out.

We thank the reviewer for the comment. As mentioned above, we changed the text accordingly and merged the paragraphs 6 and 7 (conclusion and final remarks) in one unique paragraph 6 entitled “Conclusion and future perspective”. Further, we rearranged the text pointing out two major issues that future studies need to take into account:

  • We need to better leverage the availability of new technologies to gain deeper insights into the immunological landscape after MI.
  • We need a more holistic approach implementing i) the intercommunication of different organs, ii) the cellular heterogeneity of leukocytes and iii) the stage of cardiac injury (time point after the induction of the ischemia) to further improve MI treatment.

Page 20, lines: 844-868 (highlighted text)

“In summary, recent therapeutic advances greatly helped to reduce the global burden of CVDs. However, novel insights are required to clarify yet unknown features of the pathophysiology of MI and to improve the efficacy of interventional treatment. We illustrated that the immune response to ischemic cardiac injury starts in the bone marrow and spleen - organs that are located far away from the site of tissue injury - with emergency myelopoiesis. This systemic reaction is controlled by a complex network of extracellular signals which include cytokines, chemokines, DAMPs and growth factors. Extracellular signaling impacts not only immunodynamics, but also the heterogeneity of leukocytes in different remotely located compartments and in the ischemic area. Indeed, the recent development of cell lineage tracing techniques and multiomics analysis led to a better understanding of the hematopoietic process and leukocyte heterogeneity during the course of the MI [46,164]. Thus, it is fundamental to give these findings a meaning. In other words, further studies have to clarify whether different subpopulations of myeloid cells have different roles during MI or merely represent a phenotypic shift without a particular function. Moreover, the comprehension of the signaling that shapes the cellular heterogeneity will be essential in finding new therapeutic strategies of targeting inflammation in CDVs.

The discussed pre-clinical and clinical trials greatly advanced the scientific field of cardiovascular inflammation and represent a huge step forward toward a better com-prehension of the inflammatory networks at work after MI. However, these studies do not always fully consider the heterogeneity of immune cells in hematopoietic organs and at the site of injury. New technologies, such as cell tracking and Sc-RNAseq, will greatly aid to identify novel druggable targets in MI treatment. Thus, future anti-inflammatory studies will need to carefully take into account: 1) the exact target leukocyte subpopulation, 2) when to initiate and 3) when to stop treatment after MI.”

Based on the aforementioned, I suggest minor revisions prior to the acceptance of this review.
